# Sentence Repetition Tasks to Detect and Prevent Language Difficulties: A Scoping Review

**DOI:** 10.3390/children8070578

**Published:** 2021-07-05

**Authors:** Irene Rujas, Sonia Mariscal, Eva Murillo, Miguel Lázaro

**Affiliations:** 1Departament of Psychology, Faculty of Psychology, CES Cardenal Cisneros (UCM), 28006 Madrid, Spain; 2Department of Developmental and Educational Psychology, Faculty of Psychology, Universidad Nacional de Educación a Distancia, 28040 Madrid, Spain; smariscal@psi.uned.es; 3Department of Basic Psychology, Faculty of Psychology, Universidad Autónoma de Madrid, 28049 Madrid, Spain; eva.murillo@uam.es; 4Department of Experimental Psychology, Cognitive Processes and Logopedics, Faculty of Psychology, Universidad Complutense de Madrid, 28223 Madrid, Spain; mlazar01@ucm.es

**Keywords:** early detection, sentence repetition task, sentence imitation task, early language assessment, specific language impairment, developmental language disorder

## Abstract

Sentence repetition tasks (SRTs) have been widely used in language development research for decades. In recent years, there has been increasing interest in studying performance in SRTs as a clinical marker for language impairment. What are the characteristics of SRTs? For what purposes have SRTs been used? To what extent have they been used with young children, in different languages, and with different clinical populations? In order to answer these and other questions, we conducted a scoping review. Peer reviewed studies published in indexed scientific journals (2010–2021) were analyzed. A search in different databases yielded 258 studies. Research published in languages other than English or Spanish, adult samples, dissertations, case studies, artificial models, and theoretical publications were excluded. After this exclusion, 203 studies were analyzed. Our results show that most research using SRT were conducted with English monolingual speakers older than 5 years of age; studies with bilingual participants have mostly been published since 2016; and SRTs have been used with several non-typical populations. Research suggests that they are a reliable tool for identifying language difficulties and are specifically suitable for detecting developmental language disorder.

## 1. Introduction

Sentence repetition tasks (SRTs) have been widely used in language development research for decades [1,2], with sentence repetition being part of the language batteries commonly used by clinicians to evaluate children’s language skills [3]. Despite its apparent simplicity, repeating a sentence is more than a simple memorization task. To accurately recall a sentence, participants must parse the sentence, analyze the thematic relations (i.e., order of events), interpret the underlying syntactic representation, elaborate an articulation plan, and finally, produce it [4]. It has also been argued that sentence repetition converges with comprehension data and with data from spontaneous production. In this vein, it was found that both quantitative (mean length of utterances) and qualitative measures of children’s spontaneous productions correlate with measures obtained from a repetition task in Italian [5]. Given these properties, in recent years, there has been increasing interest in studying SRT to detect and prevent language difficulties [6]. Carrying out these tasks with very young children increases the possibilities of early detection and prevention. Even though language disorders are usually diagnosed after 4 years of age, it has already been stated that there are early signs in communicative and language development that predict further difficulties [7].

In a seminal paper [8], the authors conducted a study with 11-year-old children with and without a history of developmental language impairment. The results showed that the sentence repetition task in English yielded high levels of sensitivity and specificity for diagnosing specific language impairment (SLI) in monolingual children. We are aware that the new terminology, developmental language disorder (DLD), has become increasingly accepted since 2017. However, as the term SLI has been more widely used in the studies reviewed from 2010, we have maintained it through this article. In fact, the authors observed that sentence repetition performance in recalling sentences from the Clinical Evaluation of Language Fundamentals-Revised (CELF-R) [9] was the most accurate of four clinical marker candidates of SLI; the other three markers were non-word repetition, past tense, and third person singular use. However, in a meta-analysis [10], it was concluded that although the evidence for sentence repetition as an identifier of SLI was positive, it was still inconclusive. For that research, 13 studies, undertaken with English speaking participants, were compared regarding the use of three markers for language impairment: verb tense use, non-word repetition, and sentence repetition. Sentence repetition either outperformed all other tasks or was equivalent to all other tasks in terms of identification accuracy in each of the studies that included more than one identification task. The author emphasized the need to design and carry out more studies to confirm the effects and refine the stimuli. More recently, in a community-based study [11], it was concluded that, together with an index of past tense marking in English, a sentence repetition task was sufficiently reliable for use as a language screener.

SRTs can be used not only to explore their relationship to language development in monolingual children and contribute to derive possible diagnoses, but also to explore the abilities of bilingual and multilingual children. It is important to bear in mind that, despite the fact that the majority of the world’s population is bi/multilingual, most of the phenomena related to language development have initially been carried out with monolingual samples. In this regard, it has also been suggested that SRTs have a potential advantage in second language (L2) assessment, as it has been shown that performance on this kind of task is less influenced than any other tests (e.g., standardized tests) by length of exposure to L2 and experience, which are known to be limited in L2 [12].

Regarding the construction of SRT in different languages, it is important to remember that languages can drastically differ from each other in a number of linguistic features, and this can have an impact on performance in these tasks. For this reason, cross-linguistic assessment using SRTs seems particularly relevant both in monolingual, bilingual, and multilingual participants. These kinds of tasks should be developed in different languages, according to their linguistic characteristics and particularities, and can be used afterwards with different populations. A number of studies with monolingual children who speak languages other than English, and with bi/multilingual participants with typical and non-typical language development have been carried out in recent years (see [13] for Hebrew-Russian; [14] for (European) Spanish; [15] for Welsh-English; [16] for Catalan; [17] for Arabic-German; [18] for Hungarian; [19] for Vietnamese; [20] for Czech; [21] for Cantonese; [22] for (Latino) Spanish-English; and [23] for Danish). However, drawing on the review in [13], until relatively recently, little work had focused on diagnostic accuracy of repetition tasks (SRTs and non-word repetition) in bilinguals with SLI that speak languages other than English. This situation has been compensated for in the last few years as SRTs have been developed for a European project on bilingual children with SLI within the context of a multilingual project (COST Action IS0804 “Language Impairment in a Multilingual Setting: Linguistic Pattern and the Road to Assessment” (LITMUS), http://www.bi-sli.org, accessed on 1 July 2021) [24]. Within this project, linguistically motivated sentence repetition tasks were developed for identifying bilingual children with SLI aged 5 to 8. These studies have revealed clear differences between children with SLI and typically developing (TD) children in several languages [25,26].

### Criteria to Construct, Present, and Score Sentence Repetition Tasks

In relation to the previously mentioned issue, i.e., the significant differences between languages (for instance, English is a relatively simple language concerning inflection, while Finnish is very complex), SRTs vary in the way they are constructed and may, consequently, differ in the linguistic and cognitive abilities they measure. The construction of the SRTs will differ according to the participants to be tested. Logically, depending on the age of the children to be assessed, the type and length of the sentences must be different, leading to another difficulty when comparing results across studies. For example, for school age children, [24] recommended that all sentence repetition tasks include structures that are difficult for children with SLI across languages, including wh-questions and relative clauses. Another important factor to be considered when developing a sentence repetition task is the way it is presented. In this regard, different possibilities exist, that is, the task can be presented orally or be pre-recorded and using different presentation formats (with or without visual images). Some authors suggest that although recording the items adds homogeneity, it disrupts communication between the child and the person conducting the test, while a live voice helps engage children in the task [27]. Together, all these factors add a wide range of variability in the ways the list of sentences to be repeated can be constructed and presented. 

Finally, different scoring systems can be used, from the simplest system, where the whole sentence must be repeated correctly in order to receive credit (binary scoring or 0/1), to more detailed approaches that index the number (scaled-score system) or even the types of errors per sentence. When using sentence repetition for clinical purposes, however, trade-offs may arise in using more detailed scoring systems, as simple scoring ones are faster and possibly more reliable to implement. 

Summarizing, the evidence concerning SRT is complex to address because not only were typically developing children and children with SLI tested, but also a number of other populations (e.g., attention deficit hyperactivity disorder in [28]; consistent speech/phonological disorder in [29]; resolved late talkers in [30]; and children with autism spectrum disorder and SLI in [31]). Additionally, SRTs have been used to explore monolingual, bilingual, and multilingual children with different languages as their L1 and L2. Furthermore, the tasks constructed followed different criteria depending on the age of the participants, their experimental vs. clinical use, the specific interest in particular characteristics of a given language, etc. Considering these variations, different aspects of language and cognitive processing can be at play depending on the SRT developed or used. Moreover, given this broad heterogeneity in research, drawing clear conclusions regarding the use of sentence repetition tasks is not as simple as expected. For this reason, we conducted a scoping review with the aim of shedding light on the following research questions (RQ).

Regarding languages:

RQ1: To what extent have SRTs been used in different languages?

Regarding the populations studied:

RQ2: Are these populations monolingual, bilingual, or multilingual?

RQ3: What populations have been studied using SRT?

RQ4: Can SRTs be administered to very young children (e.g., under four years of age)?

Regarding the task:

RQ4: What kinds of SRTs have been used?

Regarding the aim:

RQ5: For what purposes have SRTs been used?

## 2. Materials and Methods

We followed the guidelines of the Preferred Reporting Items for Systematic reviews and Meta-Analyses (PRISMA) statement [32] for conducting this scoping review.

### 2.1. Identification of Studies and Inclusion Criteria

The process of identifying studies for this scoping review is summarized in the PRISMA flow diagram in Figure 1.

The search was conducted in March 2021 using 11 electronic databases: Academic Search Premier; APA PsycArticles; APA PsycBooks; APA PsycInfo CINAHL Complete; EBESCO eClassics Collection (EBESCOhost); Education Source ERIC; Medline; PSICODOC; and Psychology and Behavioral Sciences.

In addition, Google Scholar was used to complete the search for studies published in 2021. 

The search was limited to peer reviewed studies published in indexed scientific journals between 2010 and 2021, in English or Spanish. Search terms included “sentence repetition task”, OR “sentence imitation task”, OR “sentence recall”. Age was limited to participants under 18.

### 2.2. Exclusion Criteria

Studies that did not meet the inclusion criteria were excluded, that is:

Theoretical studies, meta-analysis, computational modeling, case studies, dissertations and conference proceedings. 

Studies published in languages other than English or Spanish.

Studies that only included adult samples (ones that included both participants under 18 and adult samples were considered).

### 2.3. Data Analysis

The initial search led to 258 studies. After removing the 16 duplicates, 242 were screened. Thirty-nine were excluded because they did not meet the inclusion criteria. Finally, 203 full text articles were considered for the analysis. Appendix B
Table A1 lists the empirical studies included in the scoping review, and the full database can be found in the (Appendix A). For each study, we obtained the following information:

Authors and year of publication.

Journal.

Number of languages spoken by the participants: monolingual, bilingual, both, other.

Language studied.

Populations studied: typically developing children vs. non-typically developing children. 

Sample size.

Age range of the total sample included in the study.

Type of repetition task: belonging to an assessment battery, not original (taken from a previously published task), adapted (modified from a previously published task), original task (specifically developed for the particular study).

Number of sentences included in the task.

Aim of the task: we analyzed whether the SRT was used as a tool for language assessment, as a tool for cognitive assessment, as a clinical marker for language or development difficulties, or for other purposes. 

## 3. Results

### 3.1. RQ1: To What Extent Have SRTs Been Used in Different Languages?

We found 33 different languages in the studies analyzed. Half of the studies included English speaking samples (103/203) and 11% (23/203) included Spanish speaking participants. The rest of the languages appeared in less than 10% of the studies. Table 1 shows the frequency of the languages included in the studies (the total is higher than 203 because several studies include more than one language).

The percentage of studies including English remains around 50% across years. Considering the tendency towards an increase in bilingual studies, this suggests they include mostly English speaking samples as a monolingual comparison group.

### 3.2. RQ2: Are These Populations Monolingual, Bilingual, or Multilingual?

Most of the studies carried out between 2010 and 2021 included only monolingual samples (74%; 149/203). A total of 22% (45/203) included bilingual samples (13% only bilingual participants and 9% bilingual and monolingual groups). Of the studies, 1% (3/203) included other populations, mainly L2 learners, while another 2% (4/203) provided no information on the number of languages spoken by the participants.

As can be seen in Figure 2, despite the predominance of monolingual studies, the last decade has witnessed an increase in the inclusion of bilingual samples in the studies.

### 3.3. RQ3: What Populations Have Been Studied Using SRT?

Most of the studies (68%; 139/203) were carried out with children with non-typical development (NTD) with or without a TD control group, compared to 32% (64/203) of studies including only typically developing samples. For this study, we have used the term “non-typical” to cover both children with developmental disorders and children at risk of developmental difficulties (due to biological or social variables).

Of the studies, 139 included children with special characteristics. Table 2 shows the most frequent ones. As can be seen, more than half of these focused on language impairment (understanding this as a broad term covering language delay, specific language disorder, or developmental language disorder).

Sample size varies from 5 participants to 2212. However, almost 80% of the studies included samples of fewer than 150 participants, and 64% included 100 participants or less. Figure 3 shows the number of studies according to sample size.

### 3.4. RQ4: Can SRTs Be Administered to Very Young Children (e.g., under Four Years of Age)?

The age of the participants varied from 1, 10, to 25 years of age. Half of the studies (51.72%) had initial ages of between 5 and 8 years, and only 10% had initial ages of 9 years or above. Considering participants younger than 5 years, we found that 37.9% of the studies had participants aged 4 years or below. Only 15% of the studies included participants of below 4 years of age.

Regarding the final age, that is, the age of the oldest participant or at the final point of longitudinal studies, we also found that half of the studies had final ages between 5 and 8 years. Only 25% had final ages between 9 and 12 years.

As can be seen in Figure 4, we found the highest density of studies corresponded to age ranges from 5 years to around 6 years.

### 3.5. RQ4: What Kinds of SRTs Have Been Used?

Most of the studies (41%; 83/203) opted to use SRTs that had been developed and published before, that is, “not original” tests, or they partially modified previously used tests, thus being “adapted” tests. A total of 33% (68/203) of the studies used SRTs belonging to a language/cognitive assessment battery (for example, the CELF, the NEPSY, or the TOLD). Only 25% (50/203) of the studies developed original sentence repetition tests that were specifically created for the research. Two papers (1%) did not specify the SRT used in the study.

Regarding the number of sentences included in the different SRTs, the shortest one was comprised of 10, while the longest one was comprised of 180. Not all studies that used standardized tests or “not original” tasks provided information on the number of sentences included. If we only considered the SRTs that were specifically created (original) for the research, most of the studies designed 20 sentence tasks (Mode = 20), while the mean number of sentences included was 38.

### 3.6. RQ5: For What Purposes Have SRTs Been Used?

Most of the research (62%, 125/203) based on SRTs used them as a tool to assess different language abilities; 14% (28/203) used the SRT to measure cognitive abilities (for example, short-term verbal memory); and 12% (25/203) employed the SRT for other purposes (for example, to study the psychometric properties of a particular SRT or to study specific linguistic units). It is worth mentioning that 18% (36/203) of the research leveraged SRTs as a tool to identify language difficulties (clinical marker).

### 3.7. Sentence Repetition Tasks as a Clinical Marker for Language Impairment

As shown before, 36 studies were specifically undertaken to analyze the potential of sentence repetition performance to identify children with language impairment. Most of these studies (69.44%; 25/36) were designed to assess the value of SRTs used as a clinical marker for SLI; 22.22% (8/36) aimed to evaluate the potential of the SRT to identify children with language impairment, language delay, or low language abilities; 5.55% (2/36) used SRT as a clinical marker for language impairment in children with reading difficulties or dyslexia; and only two studies used SRT as a clinical marker for ASD. 

Regarding the ages of the participants included in this group of studies, mean range was 5; 4 to 8; 6 (years; months) years of age. Most of the studies were conducted with participants over 4 years of age and only two studies included children below this age. Compared to the complete set of studies reviewed, a slightly lower percentage of papers using SRT as a clinical marker included monolingual participants (66.66% compared to 74% of the whole set). Studies including bilingual populations and children with language impairment (SLI above all) have increased since 2016.

Regarding the type of task, a third of the studies (12/36) used an original format, followed by an adapted task (10/36; 7 being adaptations from the original LITMUS task to other languages), standardized batteries (9/36), and a non-original task (5/36).

Finally, in relation to the number of sentences included in the tasks, the observed range varied from 19 to 70 items (mean = 37, 26), with around 20 items being the most frequent length of SRT (mode = 20).

## 4. Discussion

The scoping review carried out with the terms “sentence repetition task”, “sentence imitation task”, and “sentence recall” revealed more than two hundred studies in the last ten years. This first result is indeed significant as it showed that SRTs have been a topic of great interest in the last decade. This is not surprising given that it is a simple task to administer, with several advantages over other language assessment tasks. For instance, as [29] highlights, SRTs enable a good number of carefully selected targets to be elicited in a more systematic way than is possible with spontaneous production. Moreover, the sentences to be repeated can include different lexical or morphosyntactic targets that are difficult to elicit with other materials or through spontaneous production. However, beyond these practical aspects, language evaluation through SRTs must be supported by experimental and empirical evidence confirming its appropriateness, for instance, for the clinical diagnosis of language impairment. In this review, we analyze evidence from the last ten years in order to offer a clear picture of the state of the field in relation to the use of different kinds of sentence repetition tasks in developmental research.

A first step towards this goal is to consider the languages under study. For this purpose, it is crucial to bear in mind that the search was limited to studies published in English (199/203) or Spanish (4/203). Considering this set of papers, data seem to be clear, with English being the most widely explored language. Spanish, being the second language in our results, was studied in more than half of the cases with Latino bilingual children (e.g., [33,34,35,36]), and research with monolingual Spanish speaking children is scarce (e.g., [14,37,38,39]). The representation of other languages is low, with most of them having only one or two published studies (e.g., in Arabic [40,41]; in Czech [20,42]; and in Kannada [43]). This set of results clearly shows that the evidence regarding the use of SRTs is biased towards English. The fact that most of the empirical evidence is related to English is important and must be considered a significant issue because differences between languages can be enormous and, therefore, in a task such as an SRT, the results for a given language do not necessarily apply to other ones. Additionally, English is a particularly simple language in terms of morphosyntax, so if we consider morphosyntactic complexity as a continuum, English can be situated at a great distance from other languages such as Finnish or Polish, which have very complex morphosyntactic structures. This is a critical trait when considering a task devoted to exploring language development, such as SRTs. Therefore, a key conclusion of this paper is that results biased towards a single language cannot represent the outcomes for other languages. This statement also holds for the studies with bilingual and multilingual participants. In these cases, English tends to be one of the participants’ languages. In fact, only 20% of the studies that included bilingual participants compared languages other than English (e.g., Arabic-German [17]; Russian-German [44]; and Spanish-Catalan [45]). In our view, it is critical that new studies with bilingual and multilingual children incorporate participants with different languages. It is true, however, that the number of studies conducted both with monolingual and bilingual children in languages other than English are slowly increasing. This, to our understanding, is of great importance, as it allows researchers to have a clearer and deeper view of the task and its value in assessing language development.

Regarding the participants involved in the research, we observed that most of the studies included children with different developmental conditions; in fact, only 32% of the studies were conducted with only TD children. Of the remaining studies (68%), most included children with SLI, but this was not the only group of non-typically developing children considered. As shown in Table 2, a considerable number of studies assess cases of children with other conditions, from cerebral palsy [46] to ASD (e.g., [47,48,49,50]). These data reflect that SRTs are not only suitable for assessing the development of children with SLI, but can also provide important information for researchers and clinicians interested in the language development of children with a number of other difficulties. Thus, these kinds of tasks have been used, for example, to explore the severity of a case of stuttering [51] or as a marker of language skills in children with dyslexia [52].

This diversity of conditions of the participants in the reviewed studies might explain, but only partially, the variability in the sample size. It is difficult to find large samples of children with rare genetic conditions, for example, but this does not hold for TD children or even for children with language disorders. The range of participants varies from 5 [53] to 2212 [54], but even if we remove these two studies, the differences are still immense (see Figure 4). Nevertheless, the important issue now is not merely the differences between studies, but the fact that the majority of them included fewer than 100 participants, with studies with groups of between 30 and 60 participants being the most numerous. It should be noted that the computation of the sample sizes reported includes all participants per study as a whole, i.e., a study on children with SLI with 60 participants will probably include only 30 children with this impairment and 30 TD children as control. This implies that despite the total number of children taking part in all these studies being large, the research is usually underpowered, and only a relatively small proportion of the papers report results from large samples. This is even more important, considering the small number of studies concerning languages other than English or specific conditions such as cerebral palsy, etc. In summary, although this review shows that SRTs have been used in research with a wide range of languages and developmental conditions, a deeper analysis indicates there is still plenty of room for more studies to be conducted.

Regarding the age of participants, most of the studies focused on children between five and eight years old. This is for two reasons. The first is that clinicians and researchers are able to engage these children in the task and obtain valid data more easily than with younger children. The second is that with a careful construction of the set of sentences to be repeated, it is possible to focus on morphosyntactic or lexical aspects of language development that are unreasonable when children are younger or older. However, our results also clearly show that the task has scarcely been used with children under 4 years of age, and less frequently with children under 3. This outcome might seem surprising, given that SRTs are devoted to assessing language development, and lower performance on the task can be used as a clinical marker for language impairment. It is a fact, however, that the administration of an SRT is complex with very young children as it is difficult for them to keep their attention focused on the task, and the results are heterogeneous and difficult to score. In any event, research has shown that is possible to conduct SRT with children under 4 years of age (see, for example, [14] for Spanish language, or [55] for English language). Nevertheless, it is an issue of major importance for clinicians and also for researchers interested in more theoretical aspects of SRTs to have data referring to children of this age. The clear gap identified in the use of sentence repetition tasks with children under 4 years old was an unexpected, but significant outcome.

Regarding the type of sentence repetition task used in the reviewed studies, 65% administered a task included in a wider battery assessment, with the CELF (in any of its versions, 4, 5, or Pre-school) being clearly the most frequently used standardized test. There are, however, a good number of other studies that developed their own set of sentences. As many as 50 created these corpora, meaning that at least, there are 50 different original tasks). Nevertheless, this number does not represent all the languages, but only part of them. Interestingly, some researchers making use of languages with little representation developed their own sets, simply because there were no previous sets to be administered (e.g., in Kannada language [43]). Therefore, although there are many languages with few studies published, most of them have their own set of sentences. In fact, the 50 studies reporting an original corpus cover 17 different languages. This number is larger than expected if we consider all the previous outcomes regarding the languages explored.

As we mentioned in the results section, not all studies that use standardized tests or “not original” tasks provide information on the number of sentences included. Considering the ones that do inform about it (see Appendix A), we found out that the number of sentences in the tasks widely varies from one to another. Examining the data, most of the studies use around 20. This number, being the mode, broadly represents the number of sentences typically used for the evaluations. Therefore, such a low number of sentences seems to be sufficiently representative of the linguistic structures needed for appropriate proper material for language assessment in every language.

Focusing on the aim of the studies, it has been stated that most of them use SRTs as a tool to assess different language abilities. In addition, 18% of the papers reviewed analyze the potential of these tasks to identify children with different language impairments, mainly children with SLI. This outcome is critical both for theoretical and clinical purposes. If a child with language problems is not identified early and does not receive the necessary intervention, behavioral and academic problems may appear. 

Even though we have not analyzed the evidence that supports the effectiveness of SRTs as a clinical marker for SLI, it seems to be well stated in the literature that the performance on SRTs contributes to the detection of children with SLI [23,56]. This is important because, as it is well known, there is currently no gold standard for the diagnosis of SLI [57] that requires the assessment of different language skills [24].

In the case of bilingual children, the number of studies using SRTs to identify children with SLI is still small and, therefore, the conclusions must be taken with caution. However, it is worth mentioning that we have observed an increase of research that include bilingual/ multilingual children in the last years, showing that SRTs are also suitable for this population. Moreover, the evidence suggests this is especially true when using the task in both the languages known by the children, as diagnostic accuracy increases in comparison to when it is administered in just one language.

For future research concerning the use of SRTs, there are still many key issues to be explored. For instance, more qualitative analyses of the results (i.e., error profiles) can be helpful to better understand the difficulties underlying SLI and also to better frame linguistic interventions. To date, few studies have addressed qualitative aspects of children´s responses, but this is a promising path for a deeper understanding of the linguistic development of children with and without typical development.

As highlighted above, future research should focus more on children under 4 years of age. More evidence is needed to ascertain whether these tasks provide useful information to detect and prevent language difficulties in young children. Should the task also prove useful for this purpose, then clinicians would have an efficient tool for assessing and guiding early intervention.

From a theoretical point of view, it remains unclear if these tasks only measure language abilities [58]. Several aspects of verbal memory, lexical knowledge, and morphosyntactic skills appear to be involved in performing the task, but more experimental designs should be carried out to obtain new data to answer the question of what the task measures.

In spite of the different needs that have been detected in this scoping review, the results in this study highlight the utility of SRTs as useful tools for assessing language abilities, and detecting and preventing language difficulties in children.

## Figures and Tables

**Figure 1 children-08-00578-f001:**
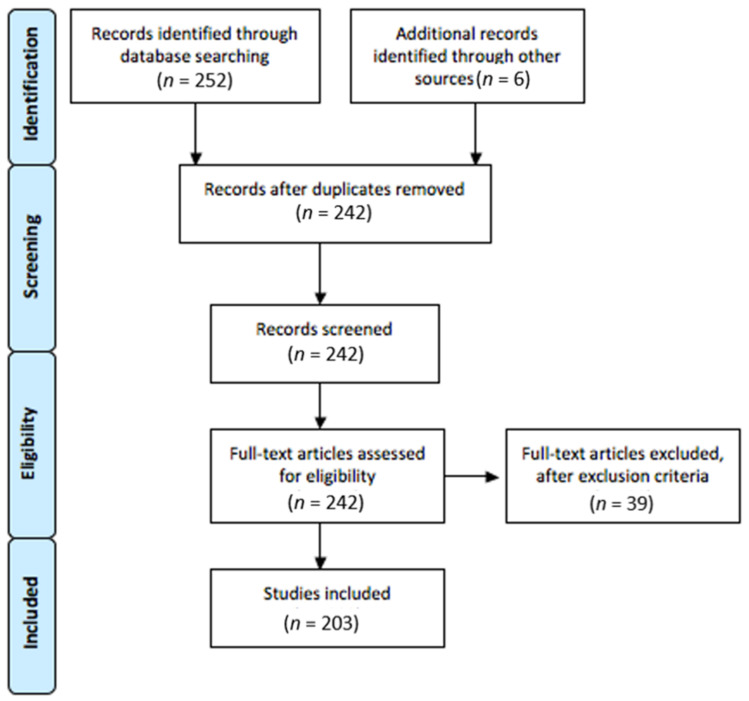
PRISMA Flow Diagram.

**Figure 2 children-08-00578-f002:**
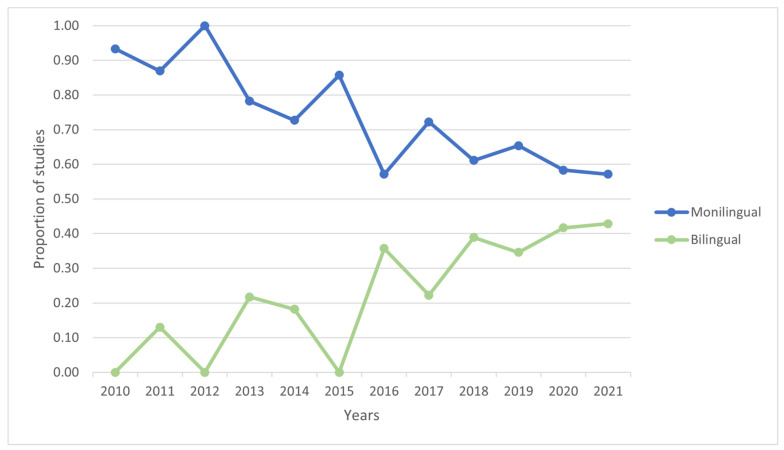
Proportion of monolingual and bilingual studies published between 2010 and 2021.

**Figure 3 children-08-00578-f003:**
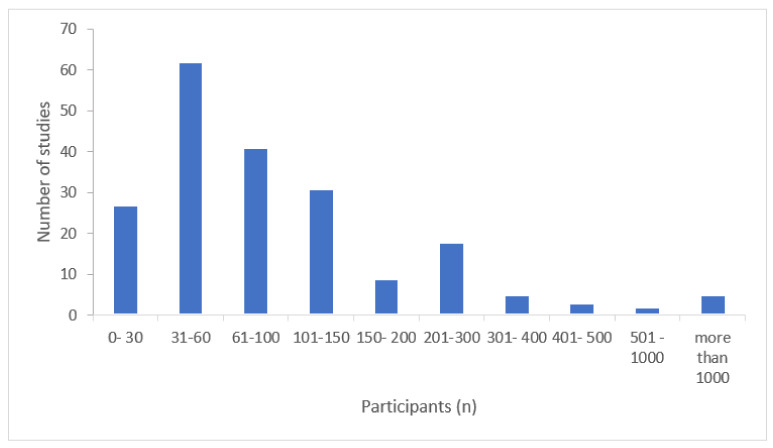
Number of studies according to the sample size.

**Figure 4 children-08-00578-f004:**
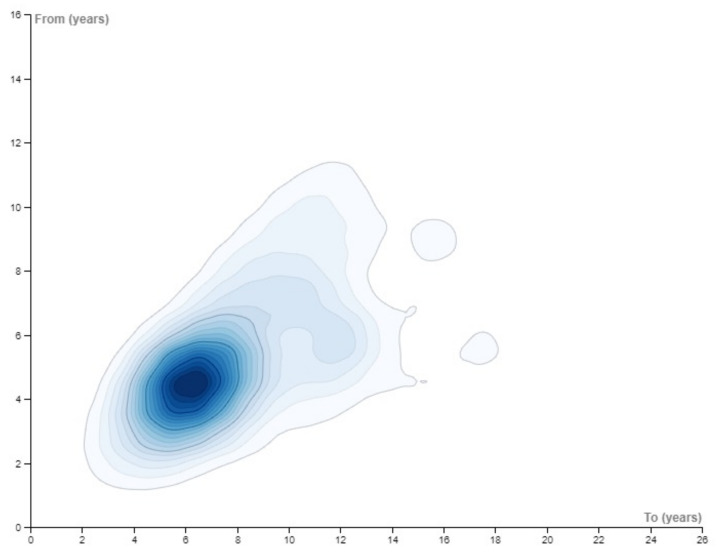
Distribution of studies according to the age range of the samples included.

**Table 1 children-08-00578-t001:** Frequency of languages included in the studies published between 2010 and 2021.

Language	*N*
English	104
Spanish	23
French	15
Italian	11
German	10
Hebrew	9
Hungarian	8
Russian	7
Arabic	6
Catalan	5
Finnish	4
Greek	4
Norwegian	4
Swedish	4
Czech	3
Danish	3
Dutch	3
Cypriot Greek	2
Kannada	2
Mandarin	2
Persian	2
Polish	2
Portuguese	2
Turkish	2
Welsh	2
Albanian-Greek	1
Cherokee	1
Farsi	1
Indian	1
Romanian	1
Vietnamese	1
British Sign Language	1
Other	1
Language Not Specified	2

**Table 2 children-08-00578-t002:** Sample characteristics studied between 2010 and 2021.

Sample Characteristics	*N*
LI or SLI or DLD or language delay	76
Deafness, hearing difficulties, or hearing loss and/or cochlear implant	14
Autism spectrum disorder	14
Children at risk for language or learning difficulties	7
Reading difficulties or dyslexia	6
Cleft palate	4
Genetic syndrome	5
Cerebral palsy or brain damage	4
ADHD	5
Speech sound disorder	5
Stuttering	2
Learning disabilities	2
Anorexia	1
Auditory processing disorders (APD)	1
Developmental coordination disorder	1
HIV-infected and HIV-exposed	1
Infantile Thiamine deficiency	1
Oncological patients	1
Pediatric bipolar disorder	1
Phonological processing deficit (PPD)	1
Preterm (very low birth weight)	1
Adopted kids	1

Note: Some studies included groups with different conditions in their samples, and thus, the total from the table is higher than 139.

## Data Availability

The full database used in the scoping review is available as Appendix A.

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
