# Peer review of "Sentence Repetition Tasks to Detect and Prevent Language Difficulties: A Scoping Review"

_children, 2021, doi:10.3390/children8070578_

Round 1

Reviewer 1 Report

Thank you for the opportunity to review this paper on sentence repetition tasks (SRTs). This is an important area to research given the impact that SRTs can have on our understanding of language development, and particularly in their ability to diagnose DLD. The authors point out that more research needs to be done on bilingual and multilingual populations and there is variability in the tasks used, so a scoping review seems appropriate. However, it is less clear why we need to focus on age (i.e. the results and discussion mention age as an important factor but there is no mention of it in the introduction) or how recently bilingual studies have been conducted. Additionally, there is no analysis of the effectiveness of SRTs as a clinical marker for DLD in the results section, which does not match the conclusions reached in the discussion. See below for more details.

Introduction

Strengths

Good overview of what SRTs are. For the most part, the introduction lays out a clear rationale for the current study, demonstrating the gap in the literature.

Limitations

However, the introduction is lacking justification for the focus on age of participants. Why is this important to study? It is also not clear why we need to know when bilingual and multilingual studies were first conducted; a finding which is highlighted in the results and discussion. More reference to the importance of ages and dates/years in the introduction would help to strengthen this argument.

p.2 lines 94-95: Can you give an example of the varied criteria used?

I think it might be beneficial for the reader if you consider labelling the research questions (RQ) RQ1, RQ2, etc. and use these same headings in the results section. There also doesn’t appear to be a question about the different formats despite the literature discussed in the introduction about different scoring systems.

Method

The method is brief, but clear. However, this paper is missing appendix A and the supplementary materials (excel spreadsheet is not readable) so I cannot fully comment on this section.

Results

Strengths

Generally this section was clear and I appreciated the presentation of results in interesting figures. 

Limitations

As before, the headings in the results section match up with the information listed in the “data analysis” section on p.4/5 but this does not line up with the RQs. I think it would help with consistency to rename these sections in line with RQs.

p.6 line 207-209 Is this true to say? Could it also be due to the fact that for many bilingual speakers their second language is English?

p.7 Table 2 I think it is incorrect to label being adopted as a “condition”. Was being adopted classified as “non-typical” development in the original paper? I don’t think this should be included or the table should be renamed. Perhaps “sample demographic” or something similar would be better.

p.8 Type of task: Why is there no mention of the different types of scoring here? There is a lot more detail about the type of task in section 3.9 – does the type of task only matter for diagnosing DLD?

p.8 Number of sentences: Would be interesting to include the mode/mean number of sentences used in the “not original” SRTs and the SRTs from assessment batteries to see the variance between ‘standardised’ tasks and original tests.

Discussion

Strengths

There are some insightful conclusions drawn about the need for more studies of different languages, the underpowering of studies due to small sample sizes and an interesting point about the original SRTs designed for different languages.

Limitations

There are some major limitations with the discussion given that there are some conclusions drawn about the effectiveness of SRTs as a clinical marker for DLD that are not analysed in the results. Also, the discussion does not close with a strong, final conclusion as it mentions uncertainty about what the SRT actually measures. More details are given below:

p.11 line 399-401: “Therefore, such a low number of sentences must be sufficiently representative of the 399 linguistic structures needed for appropriate proper material for language assessment in 400 every language.” Can this be concluded given that we don’t have a comparison to the “non-original” tasks?

p.11 line 407-410: This was not stated in the results. Perhaps inclusion of the effectiveness of SRTs as a clinical marker would be more impactful in the results, instead of simply summarising the number of papers who investigated this.

p.12 line 416 onwards: Again, these conclusions do not appear to flow from the findings presented in the results section. Inclusion of the effectiveness of SRTs as a clinical marker would definitely help.

Minor spelling/grammar

p.6 line 216: “abroad” should be “a broad”

p.9 line 302: rephrase “…state of the art…” as it usually means futuristic. Perhaps change to “…state of the field…”?

p.11, line 385: remove second “being”

Author Response

Dear reviewer,

We appreciate your careful reading and time invested in our study. We have worked on all the queries raised, as follows. Changes are highlighted in red in the text.

We hope the new version suits fine.

Best,

The authors.

Reviewer 2 Report

This is a well written scoping review aiming at collecting evidence on the sentence repetition paradigm in language acquisition.

I have truly enjoyed reading the review and have no comments to ask to the authors. Well done for all your hard work and thanks for this contribution.

Author Response

Dear reviewer,

Thank you very much for your acknowledgment and support.

We have worked on some comments from other reviewers and we hope the new version suits fine.

Best,

The authors.

Reviewer 3 Report

I inserted some comments in the manuscript. Minor revisions.

Author Response

Dear reviewer,

We appreciate your careful reading and time invested in our study. We have worked on all the queries raised. Changes are highlighted in red in the text.

We hope the new version suits fine.

Best,

The authors.
